# Serum Levels of Dihomo-Gamma (γ)-Linolenic Acid (DGLA) Are Inversely Associated with Linoleic Acid and Total Death in Elderly Patients with a Recent Myocardial Infarction

**DOI:** 10.3390/nu13103475

**Published:** 2021-09-30

**Authors:** Dennis Winston T. Nilsen, Peder Langeland Myhre, Are Kalstad, Erik Berg Schmidt, Harald Arnesen, Ingebjørg Seljeflot

**Affiliations:** 1Department of Cardiology, Stavanger University Hospital, 4068 Stavanger, Norway; 2Department of Clinical Science, Faculty of Medicine, University of Bergen, 5020 Bergen, Norway; 3Institute of Clinical Medicine, Faculty of Medicine, University of Oslo, 0315 Oslo, Norway; p.l.myhre@medisin.uio.no (P.L.M.); are.kalstad@medisin.uio.no (A.K.); UXHAAR@ous-hf.no (H.A.); UXINLJ@ous-hf.no (I.S.); 4Department of Cardiology, Division of Medicine, Akershus University Hospital, 1474 Lørenskog, Norway; 5Center for Clinical Heart Research, Department of Cardiology, Oslo University Hospital Ullevaal, 0424 Oslo, Norway; 6Department of Cardiology, Aalborg University Hospital, 9000 Aalborg, Denmark; ebs@dcm.aau.dk

**Keywords:** *n*-6 fatty acids, dihomo-gamma (γ)-linolenic acid, DGLA, LA:DGLA ratio, delta-6-desaturase, biomarker, prognosis, myocardial infarction, all-cause mortality

## Abstract

Dihomo-gamma-linolenic acid (DGLA) is an *n*-6 polyunsaturated fatty acid (PUFA) derived from linoleic acid (LA). The LA:DGLA ratio reflects conversion from LA to DGLA. Low levels of DGLA in serum have been related to poor outcome in myocardial infarction (MI) patients. **Aims:** To assess the association of DGLA and LA:DGLA with total death as a primary aim and incident cardiovascular events as a secondary objective. **Methods:** Baseline samples from 1002 patients, aged 70 to 82 years, included 2–8 weeks after an MI and followed for 2 years, were used. Major adverse clinical events (MACE) consisted of nonfatal MI, unscheduled coronary revascularization, stroke, hospitalization for heart failure or all-cause death. Cox regression analysis was used to relate serum *n*-6 PUFA phospholipid levels (%wt) to the risk of MACE, adjusting for the following: (1) age, sex and body mass index (BMI); (2) adding baseline cod liver oil supplementation; (3) adding prevalent hypertension, chronic kidney disease and diabetes mellitus. **Results:** Median DGLA level in serum phospholipids was 2.89 (Q1–Q3 2.43–3.38) %wt. DGLA was inversely related to LA and LA:DGLA ratio. There were 208 incident cases of MACE and 55 deaths. In the multivariable analysis, the hazard ratio (HR) for the total death in the three higher quartiles (Q2–4) of DGLA as compared to Q1 was 0.54 (0.31–0.95), with *p* = 0.03 (Model-1), 0.50 (0.28–0.91), with *p* = 0.02 (Model-2), and 0.47 (0.26–0.84), with *p* = 0.012 (Model-3), and non-significant for MACE. Risk of MACE (Model 3) approached borderline significance for LA:DGLA in Q2–4 vs. Q1 [HR 1.42 (1.00–2.04), *p* = 0.052]. **Conclusions:** Low levels of DGLA were related to a high LA:DGLA ratio and risk of total death in elderly patients with recent MI.

## 1. Introduction

Studies in primary and secondary prevention, replacing saturated fatty acids with polyunsaturated fatty acids (PUFA’s) in the diet, have been burdened with design limitations, and whether increased intake of linoleic acid (LA) and its derivatives may provide a health benefit is, therefore, still being questioned [1]. However, a recent meta-analysis supports a favorable role for LA in cardiovascular disease (CVD) prevention [2].

LA is an essential *n*-6 PUFA and is primarily derived from vegetable oils, such as corn, sunflower, safflower and soy [1]. Its intake has increased from 1 to 2% before the 1930s, to more than 7% of daily calories, corresponding with an average daily ingestion of 15 g in Western populations [1,3]. In line with this, the ratio between *n*-6 and *n*-3 PUFA has changed from 4:1 to 20:1 [3,4]. As shown in Figure 1, LA (18:2*n*-6) can be desaturated to gamma-linolenic acid (GLA; 18:3*n*-6), which in turn is elongated into dihomo-gamma-linolenic acid (DGLA; 20:3*n*-6), from which arachidonic acid (AA; 20:4*n*-6) may be formed upon further desaturation [5]. Despite the increased consumption of LA, its elongation-desaturation product, AA, and products of the essential *n*-3 PUFA, have remained relatively constant at <1% energy since the early 1900s [3].

Previous reports suggest a cardioprotective effect of high intakes of LA [2,6] and high packed erythrocyte concentrations of LA [7], and AA in a meta-analysis [8] was associated with a lower risk of CVD. The degree of conversion of LA to AA is very low and about 0.2% [9]. Approximately 0.15 g of AA is ingested per day from meat, eggs and some fish [1], whereas GLA and DGLA are consumed in even smaller amounts and their levels are nearly exclusively derived from metabolism from LA.

The first step in the metabolism of LA (Figure 1) is by delta-6-desaturation to gamma-linolenic acid (GLA). This is a slow and rate-limiting process, whereas the elongation to DGLA by delta-6-elongase is rapid. Thus, conversion to DGLA can be expressed by the ratio of LA to DGLA (LA:DGLA) [10]. The generation of DGLA is inversely associated with age, due to a decline in the rate-limiting step of delta-6 desaturation [11].

In addition to being a precursor of AA, DGLA may display anti-inflammatory and antiproliferative properties [9]. Cyclooxygenase (COX) 1 and 2 convert DGLA into prostaglandins of the 1-series (PGE-1), and the action of 15-lipoxygenase (LOX) provides 15-(S)-hydroxy- 8,11,13-eicosatrenoic acid (15-HETrE) [12,13]. These metabolic products of DGLA are generally considered to have anti-inflammatory, antiproliferative and antithrombotic properties, whereas the 2-series prostaglandins (PGE2) and leukotrienes derived from AA may promote inflammation and thrombosis [5,13,14,15]. This is illustrated in Figure 1.

Smaller studies have investigated the biochemical effects of supplementation with GLA, which is rapidly converted to DGLA by the action of elongase-6, increasing plasma levels of DGLA [16,17]. It is, however, still unknown to what extent downstream metabolic products of DGLA are further formed. That also applies to their relevance to CVD. However, as a first step, risk assessment according to DGLA levels may lead to further insight.

In a previous prospective observational study of patients presenting with an acute coronary syndrome, higher concentrations of DGLA in packed red blood cells were independently associated with reduced all-cause mortality [HR 0.55 (95% CI, 0.35–0.88), *p* = 0.012] during a median of 7 years of follow-up [18].

To further explore whether DGLA or other *n*-6 PUFAs provide prognostic information, we have studied their utility as predictors of outcome, when measured in serum phospholipids obtained from patients included in the OMEMI (Omega-3 Fatty acids in Elderly with Myocardial Infarction) trial [19,20]. Furthermore, we have related the LA:DGLA ratio, reflecting delta-6-desaturase activity, to serum levels of DGLA and to clinical outcomes.

## 2. Methods

The OMEMI trial [19] was a multicenter, placebo-controlled, double-blind clinical trial, evaluating the effect of a daily intake of 1.8 g *n*-3 PUFA vs. a corresponding amount of corn oil (placebo) on combined major adverse clinical events (MACE), i.e., myocardial infarction (MI), unscheduled coronary revascularization, stroke, hospitalization for heart failure or all-cause death in elderly post-MI patients. In total, 1027 participants aged 70–82 were included 2 to 8 weeks after their index MI. The study was organized by the Center for Clinical Heart Research, Department of Cardiology, Oslo University Hospital, Oslo, Norway, and was conducted by independent investigators at four Norwegian hospitals according to Good Clinical Practice (GCP) rules and in compliance with the declaration of Helsinki, following the approval by the Regional Committee for Medical and Health Research Ethics. All participants provided written informed consent and the trial was registered at ClinicalTrials.gov (NCT01841944), posted 29 April 2013. Design and methods have previously been published [20]. Mean follow-up was 313 ± 207 days. Patients were seen by a study physician at baseline visit and after 3, 12 and 24 months, or contacted by telephone if they were unable to attend. All information was recorded in a Case Report Form (CRF) and adverse events were assessed by a Data and Safety Monitoring Board (DSMB). One child spoon of cod liver oil was allowed as continuation of a regular supplementation, used by many participants. A flow chart is shown in Appendix A. Blood samples were available from 1002 patients and were obtained in fasting state between 8:00 and 11:30 a.m. Serum was prepared and frozen at −80 °C for analyses of fatty acid composition in serum phospholipids, performed at the Lipid Research Laboratory (Aalborg University Hospital, Aalborg, Denmark) by gas chromatography. The relative proportion of each fatty acid was expressed as percent weight (%wt) of total fatty acids [19,21,22]. The coefficients of variation for the relevant fatty acids were: LA 0.40%, DGLA 0.75%, GLA 5.37% and AA 0.60%.

## 3. Statistical Analyses

Normally distributed continuous variables are given as means ± SD. Skewed variables are given as median values with interquartile range (IQR). ANOVA was used for normally distributed continuous variables, the Kruskal–Wallis test for non-normally distributed variables and the chi-square test for categorical variables. Clinical characteristics are presented by quartiles of baseline DGLA concentrations and quartiles of LA:DGLA ratio, respectively, and compared for trend across quartiles (Q) by linear and logistic regression. Continuous values and quartiles of predictor variables at inclusion were subjected to Cox regression analyses for time-to-event analysis. Treatment interactions, as well as violation of the proportional hazard assumption were tested in relation to outcome. A cubic spline was introduced to show the relationship between DGLA and total death. A set of potentially confounding variables was selected a priori and included in the adjusted regression models. Three models were employed in the multivariable analysis, correcting for: (1) age, sex and body mass index (BMI); (2) adding regular intake of cod liver oil at baseline; (3) adding hypertension, chronic kidney disease and diabetes mellitus. We also tested whether DGLA and LA:DGLA ratio, respectively, were treated independent of circulating lipids that were not adjusted for in the multivariable analysis.

The hazard ratio (HR) according to quartiles relative to the lowest quartile is presented with the 95% confidence interval (CI). The chi-square test was applied when comparing proportions for qualitative variables among quartiles at baseline. The statistical analyses were performed with Stata Software (version 16, Stata Corp., College Station, TX, USA). Tests were applied with a two-sided significance level of 5%.

## 4. Results

As the overall OMEMI study [19] was neutral, we used the entire population to assess the prognostic utility of *n*-6 PUFA levels in relation to clinical outcome, after testing for treatment interaction in relation to MACE (*p* = 0.40).

Serum levels of LA, GLA, DGLA and AA were available in 1002 of the 1027 patients. The total number of the composite endpoint of first occurring event (MACE) was 208, which included 40 deaths as the first event. The total number of non-survivors was 55. The baseline distribution of DGLA measured in serum phospholipids is shown in Appendix A and the median value and range of DGLA was 2.89 (IQR 2.43–3.38) %wt.

Baseline characteristics related to DGLA quartiles are given in Table 1. Age was slightly higher in the lowest quartile of DGLA, but less than one year compared to the other quartiles (*p* = 0.019). There was a gradual increase in BMI (*p* < 0.001) with increasing quartiles of DGLA, and a higher proportion of hypertensive subjects in the highest as compared to the lowest quartile (*p* = 0.05).

HDL-cholesterol concentrations were lower and triglyceride concentrations higher (*p* < 0.001 for both) with increasing quartiles of DGLA, whereas LDL-cholesterol concentrations were rather similar (*p* = 0.35). LA was inversely and significantly (*p* < 0.001) related to DGLA, whereas GLA, although to a modest extent, showed a parallel relationship to the content of DGLA in serum phospholipids (*p* < 0.001). There was a steady decrease in the LA:DGLA ratio across the quartiles of DGLA, with a ratio twice as high in the lowest than in the highest quartile of DGLA.

The distribution of variables through the quartiles of the LA:DGLA ratio was inversely related to that of the DGLA quartiles (Table 2).

After testing for violation of the proportional hazards assumption for DGLA in relation to MACE (*p* = 0.14), further statistical analysis was performed.

### 4.1. DGLA as Predictor of Adverse Outcome

Table 3 shows the distribution of endpoints across quartiles of DGLA. Statistically, no significant differences were observed, but there were almost twice as many total deaths in the lowest as compared to the higher quartiles, with a borderline trend across quartiles (*p* = 0.06).

Univariate analysis of the prognostic utility of continuous DGLA values for MACE revealed an HR of 0.89 (0.72–1.08) per %wt increase in DGLA, *p* = 0.24. When comparing Q2–4 to Q1, an HR of 0.76 (0.56–1.02), *p* = 0.067, was obtained. These associations remained essentially unchanged and statistically non-significant in the multivariable Cox regression models (Table 4).

Univariate analysis of the prognostic utility of continuous DGLA values for total death revealed an HR of 0.72 (0.48–1.07) per %wt increase in DGLA, *p* = 0.11, whereas an HR of 0.51 (0.30–0.88), *p* = 0.016, was obtained when comparing Q2–4 with Q1 of DGLA. A restricted cubic spline showing the relationship between DGLA (*x*-axis) and all-cause mortality (*y*-axis) is displayed in Appendix A.

DGLA remained an independent predictor of total death after adjusting for potential confounders in Model 1, 2 and 3, as shown in Table 4, with an HR in Model 3 of 0.47 (0.26–0.84), *p* = 0.012.

The Kaplan–Meier plot of time to death and MACE within 2 years by quartiles of the DGLA ratio are given in Figure 2.

### 4.2. LA:DGLA Ratio as Predictor of Outcomes

Events by quartiles of the LA:DGLA ratio are shown in Table 5. There was an increase in number of events from Q1 to Q4 (*p* = 0.049), mainly driven by stroke (*p* = 0.05) and total death (*p* = 0.09).

The univariate association for MACE in Q4 of LA:DGLA compared to the lowest quartile (Q1) was HR 1.49 (1.00–2.23), *p* = 0.005, but after fully adjustment in Model 3, the HR for Q4 was weakened [1.45 (0.95–2.22), *p* = 0.083]. The HR for Q2–4 combined as compared to Q1 was significant in the univariate analysis [1.42 (1.09–1.99), *p* = 0.046], but became borderline significant after adjusting for the variables in Model 3 [1.42 (1.00–2.04), *p* = 0.052], as shown in Table 4.

The number of total deaths were more than two-fold higher in Q4 as compared to Q1 [HR 2.30 (1.04–5.06), *p* = 0.038], and became borderline significant in the fully adjusted model [HR 2.39 (1.00–5.77), *p* = 0.051]. When comparing Q2–4 combined to Q1 after adjusting for the variables in Model 3, statistical significance was not obtained [HR 1.80 (0.83–3.92), *p* = 0.14].

The Kaplan–Meier plot of time to death and MACE within 2 years by quartiles of LA:DGLA ratio are shown in Figure 3.

### 4.3. LA, GLA and AA as Predictors of Total Death and MACE

The distribution of events through quartiles of LA, GLA and AA is shown in Appendix A, respectively. Serum phospholipid levels of LA were not associated with outcome, whereas the number of strokes showed a gradual and significant reduction from the lowest to the highest quartile of GLA (*p* = 0.005), more pronounced than the trend obtained across the DGLA quartiles. The univariate HR for continuous AA values in relation to MACE and total death was 0.89 (0.72–1.08), with *p* = 0.24, and 0.96 (0.85–1.08), with *p* = 0.48, respectively.

## 5. Discussion

In elderly patients with a recent MI, low baseline values of DGLA in serum phospholipids were independently associated with increased mortality, after adjusting for age, sex, BMI, baseline *n*-3 supplementation, hypertension, chronic kidney disease and diabetes mellitus. Baseline concentrations of the other measured *n*-6 PUFAs (LA, GLA and AA) were not associated with clinical endpoints.

Our findings related to total deaths are in line with the results obtained in the Risk Markers of Acute Coronary Syndromes (RACS) registry, a 7-year follow-up of predominantly older patients experiencing symptoms consistent with an acute coronary syndrome [18]. In that study, DGLA was measured in red blood cells, and was found to independently predict total mortality, and a combination of total death, MI or stroke.

In the present OMEMI substudy, low levels of DGLA independently predicted all-cause mortality, but we were not able to demonstrate an association with the predefined MACE in the main OMEMI study [19], as previously noted in the RACS registry [18]. This may be due to a shorter follow-up time and fewer cardiovascular events in the OMEMI study, despite the fact that OMEMI included 2.5 times as many patients. However, baseline characteristics differ substantially between the two studies with respect to prophylactic vascular medical treatment, and furthermore, more than twice as many patients in the OMEMI trial had been revascularized prior to inclusion.

Our results are also largely consistent with those of Ouchi et al. [23], who recently found an association between serum PUFA levels and long-term mortality. In that study, DGLA and AA levels were significantly lower in non-survivors as compared to survivors. In contrast, AA did not predict total death in the present OMEMI substudy.

Members of the *n*-6 series of FAs did not affect variables related to biochemistry and blood pressure in a meta-analysis of four randomized primary prevention studies, which included 660 individuals followed for 24 weeks [24]. Moreover, Harris et al. [25] examined cardiovascular outcomes and death after a median of 7 years of follow-up of 2500 individuals with a mean age of 66 years in the Framingham Heart Study Offspring cohort without prevalent CVD, and found no association between the content of *n*-6 FAs in erythrocytes and total deaths, and no association with cardiovascular events. Reasons for heterogeneity in the results between these studies may be the clinical setting (primary versus secondary prevention) and age difference, and perhaps, also different diets.

Randomized, controlled studies in primary and secondary prophylaxis, replacing saturated fatty acids with PUFAs in the diet, have been burdened with design limitations, such as lack of blinding and ambiguity related to whether addition of one or removal of another fatty acid constituent may have exerted an effect [1]. However, a meta-analysis [6] including six such trials indicated that PUFAs may lower the risk of coronary events by 24% in patients with established coronary heart disease.

Although DGLA levels increase by GLA supplementation [16,17], a more recent review of the literature [26], focusing on cardiovascular effects of randomized *n*-6 FA intervention in both primary and secondary prevention, did not reveal any clinical benefit of these fatty acids, except for the occurrence of MI. Thus, no benefit was noted for all-cause mortality, cardiovascular (CVD) mortality, CVD events and risk factors (blood lipids, adiposity, blood pressure). This meta-analysis [26] included 19 randomized trials with a total of 6461 participants with or without CVD who were followed for one to eight years. Seven of the included trials in this meta-analysis assessed the effects of supplemental GLA and twelve of LA, whereas none included supplements with DGLA or AA. However, hypothetically, one can assume that DGLA would be present in sufficient quantities to exert a beneficial effect in those studies in which subjects were treated with GLA supplements. However, the number of participants was low and a meta-analysis with this cohort size would not have the power to show any clinical benefit on cardiovascular endpoints.

Contrary to previous observations in red blood cells [18], where LA showed a non-significant trend in the same direction as DGLA, we found that in serum phospholipids, levels of LA were inversely related to DGLA, and did not confer any prognostic information in relation to MACE and total death in the OMEMI substudy. The metabolic pattern of these fatty acids may differ in serum phospholipids as compared to that of red blood cells, in which they are incorporated into cellular membranes, rendering them less interchangeable and less exposed to enzymatic turnover. In serum phospholipids, the inverse relationship between DGLA and LA may better express delta-6-desaturase activity.

As described in further detail by Fan and Chapkin [14], LA is metabolized in a variety of tissues by delta-6-desaturase to form GLA, which rapidly elongates by the action of delta-6-elongase to DGLA, as demonstrated by the rapid turnover of supplements of GLA to DGLA, by which delta-6-desaturase is bypassed [16]. This is consistent with our findings showing very low levels of GLA as compared to DGLA.

As reduced levels of DGLA may result from reduced enzymatic conversion of its precursor LA, the major *n*-6 PUFA in the diet, an increase in the ratio of LA to DGLA (LA:DGLA) has been shown to reflect reduced delta-6-desaturase activity [11], the rate-limiting enzyme responsible for the conversion from LA to gamma linolenic acid (GLA). In the present study, we found that high levels of LA:DGLA ratio were statistically significantly associated with MACE in univariate analysis, with borderline significance after adjustment in the multivariable analysis, whereas statistical significance was not obtained when relating LA:DGLA to total mortality. However, our findings suggest that reduced delta-6-desaturase activity may be responsible for low DGLA levels. This may result in a reduction in levels of the 1-series of thromboxanes and prostanoids, and the 15-hydroxyl derivative that counteract the 2-series of prostanoids and leukotrienes, respectively, derived from AA [9,13].

Overall, our results point in the opposite direction to those obtained by Warensjø et al. [27], who noted an increased risk of mortality associated with a high estimated delta-6-desaturase activity in a community-based prospective sample of 50-year-old men followed for a maximum of 33.7 years. Interestingly, as for LA, results obtained for DGLA and delta-6-desaturase in secondary prevention do not seem to apply in primary prevention.

In the study by Warensjø et al. [27], an inverse relation between delta-5-desaturase and mortality was also noted. As the level of AA is mainly determined by diet and to a lesser degree dependent on the level of its precursors [3], estimating the conversion of DGLA to AA was not an issue in the current study. However, the importance of the counteracting effects of DGLA and its derivatives to the unfavorable metabolites of AA should be kept in mind. According to Johnson et al. [16], DGLA but not AA, accumulates in neutrophil glycerolipids after 3 and 6 g daily GLA supplementation, suggesting that the increase in DGLA relative to AA within inflammatory cells may represent a mechanism by which dietary GLA exerts an anti-inflammatory effect.

Focusing on DGLA as a risk marker in patients with established CVD, there are several issues that need to be discussed. It has been shown that GLA supplementation is associated with minor conversion to AA, resulting in an accumulation of DGLA [14,16]. Furthermore, as preliminary data [24] indicate that high DGLA levels after GLA supplementation do not improve CVD outcome, it is tempting to postulate a threshold level beyond which DGLA will not exert additional clinical benefits. Its place as a risk marker may be limited to a nutritional range below this threshold, as shown by the distribution of events through the quartiles of DGLA in our study, in line with that of *n*-3 fatty acids [28].

It may not only be DGLA per se, but rather its role as a precursor for anti-inflammatory, antiproliferative [13] and antithrombotic substances [15] that may favorably influence prognosis. As minor amounts of DGLA are converted to AA during GLA supplementation, the accumulation of DGLA may have exceeded the capacity to further increase the formation of physiologically beneficial substances, consisting of prostaglandins of the 1-series (PGE-1) and 15-(S)-hydroxy-8,11,13-eicosatrenoic acid (15-HETrE). This would theoretically also support a threshold related to a beneficial effect of DGLA.

As the conversion of DGLA to AA is of less importance as compared to AA provided in the diet, an additional ratio between DGLA and AA would not be a reliable measure for delta-5-desaturation. Furthermore, looking at AA separately did not confer prognostic information, neither did LA, despite an inverse linear relationship with DGLA, and our current data would infer a prognostic potential essentially related to DGLA (and GLA), and ultimately to delta-6-desaturase.

### Limitations

Our data would suggest an association between reduced levels of delta-6-desaturase activity and MACE, which is largely driven by all-cause mortality, mostly of cardiovascular origin.

The LA:DGLA ratio will to some degree reflect the dietary intake of LA in addition to enzymatic conversion. Furthermore, the risk related to the LA:DGLA ratio may also depend on a threshold, as observed for DGLA. Therefore, this ratio was analyzed in quartiles and not as a continuous variable.

It may be argued that the ratio of LA to GLA or vice versa [29] may serve as a better indicator for delta-6-desaturase activity. However, levels of GLA are more than 30 times lower than levels of DGLA, with a CV of 5.37% as compared to 0.75% for DGLA, and also taking into account that GLA is rapidly elongated to DGLA, the latter was chosen as a denominator, to ensure reliable reproducibility.

Referring to AA, we cannot preclude the possibility that the high median combined baseline levels of eicosapentenoic acid (EPA) and docoshexaenoic acid (DHA) in the RACS registry (6.4% in packed red blood cells) [18] and in serum phospholipids in the OMEMI trial (8.5%), respectively, may have attenuated an association of AA with outcome. However, high levels of AA were not associated with higher cardiovascular mortality, incident CVD, incident CHD or incident stroke in a pooled biomarker analysis of 30 prospective studies [2].

In our multivariable model, we adjusted for BMI, as information on waist circumference, which is considered to be more relevant to survival and CVD outcomes [30], was not available. We also adjusted for codliver oil supplementation at baseline, as there was a gradual decrease in the use of fish oil supplementation across the quartiles, from Q1 to Q4, *p* for trend <0.001. Moreover, we have no information related to lifestyle and genetic variations of fatty acid desaturases. We also adjusted for age, sex, hypertension, chronic kidney disease and diabetes mellitus. Whereas the total number of patients in our study was high as compared to similar studies, the number of total deaths (*n* = 55) was low.

## 6. Conclusions

In this OMEMI substudy, low levels of serum phospholipid DGLA were associated with increased risk of total death in elderly patients with a recent MI. Further studies are needed to confirm and extend our findings.

## Figures and Tables

**Figure 1 nutrients-13-03475-f001:**
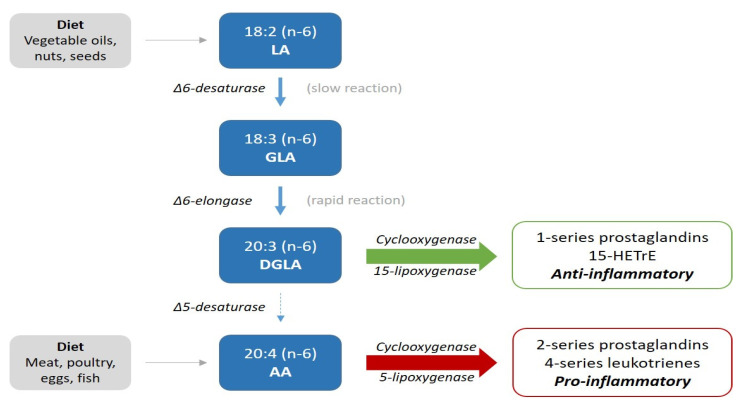
Illustration of the metabolic pathway of *n*-6 fatty acids.

**Figure 2 nutrients-13-03475-f002:**
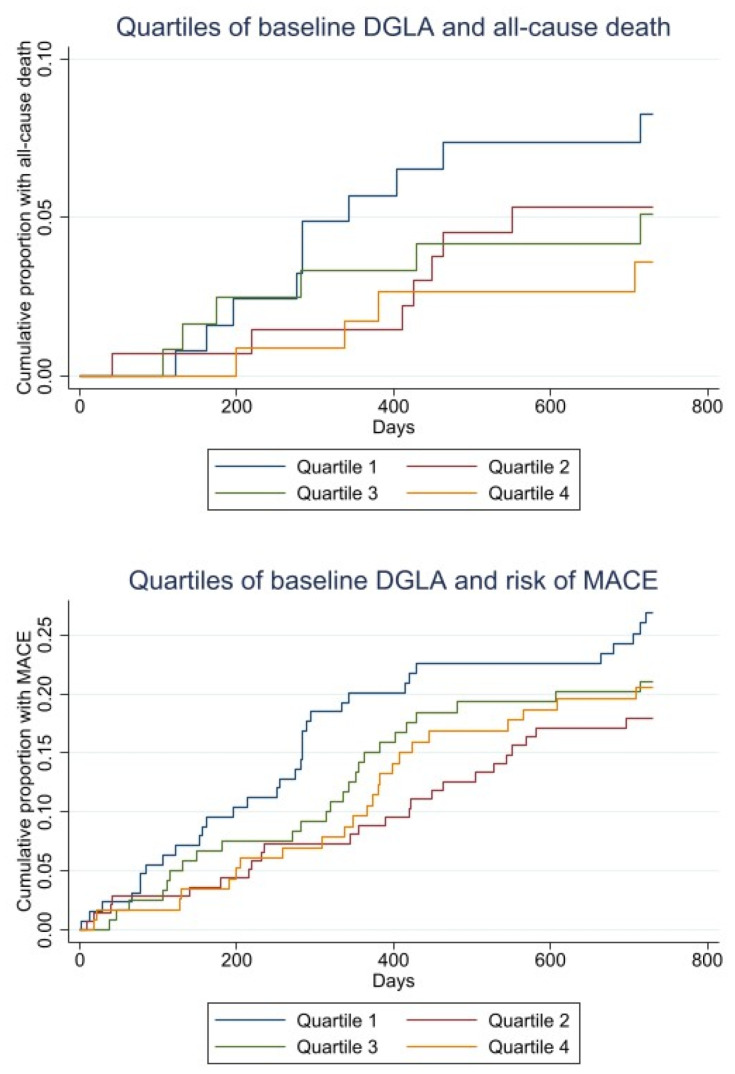
Kaplan–Meier curves for baseline quartiles of DGLA and all-cause death (upper). Kaplan–Meier curves for baseline quartiles DGLA and risk of MACE (lower).

**Figure 3 nutrients-13-03475-f003:**
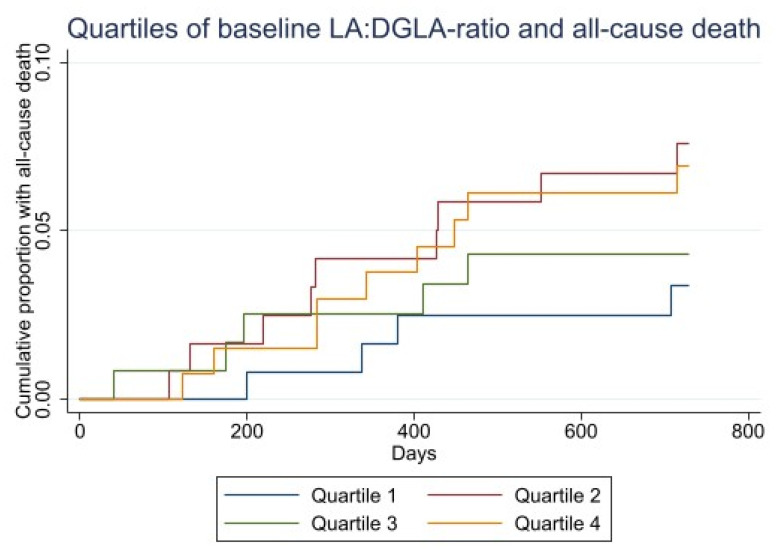
Kaplan–Meier curves for baseline LA:DGLA quartiles and all-cause death (upper). Kaplan–Meier curves for baseline LA:DGLA and risk of MACE (lower).

**Table 1 nutrients-13-03475-t001:** Baseline characteristics by serum phospholipid quartiles (Q) of DGLA.

	DGLA	DGLA	DGLA	DGLA Q 4	*p*-Value for Trend
	Q1	Q2	Q3	Q4	
No. of Patients	*n* = 251	*n* = 251	*n* = 251	*n* = 249	
*DGLA, %wt*	*2.1 ± 0.3*	*2.7 ± 0.1*	*3.1 ± 0.1*	*3.8 ± 0.4*	
*DGLA, %wt range*	*0.90–2.43*	*2.44–2.89*	*2.90–3.38*	*3.39–5.84*	
Age, years	75.3 ± 3.7	74.4 ± 3.5	74.9 ± 3.6	74.5 ± 3.5	0.019
Sex (females) *n* (%)	73 (29.1%)	73 (29.1%)	68 (27.1%)	73 (29.3%)	0.94
Smoking history					0.17
Current smoker *n* (%)	27 (10.8%)	41 (16.3%)	25 (10.0%)	25 (10.0%)	
Previous smoker *n* (%)	118 (47.0%)	122 (48.6%)	131 (52.2%)	120 (48.2%)	
Never smoker *n* (%)	106 (42.2%)	88 (35.1%)	95 (37.8%)	104 (41.8%)	
BMI kg/m^2^	25.6 ± 3.8	26.1 ± 3.7	27.7 ± 9.8	27.6 ± 4.4	<0.001
SBP mmHg	138.2 ± 19.8	136.7 ± 20.1	136.7 ± 19.1	136.8 ± 20.2	0.78
LVEF %	49.0 ± 8.6	50.8 ± 8.7	50.3 ± 8.4	50.0 ± 8.3	0.29
Diabetes mellitus *n* (%)	56 (22.3%)	44 (17.5%)	52 (20.7%)	56 (22.5%)	0.49
Hyperlipidemia *n* (%)	114 (45.4%)	120 (47.8%)	120 (47.8%)	109 (43.8%)	0.76
Hypertension *n* (%)	144 (57.4%)	152 (60.6%)	140 (55.8%)	167 (67.1%)	0.05
CKD *n* (%)	10 (4.0%)	11 (4.4%)	12 (4.8%)	12 (4.8%)	0.97
Heart failure *n* (%)	13 (5.2%)	18 (7.2%)	17 (6.8%)	16 (6.4%)	0.82
LDL chol. mmol/L	2.0 ± 0.7	2.0 ± 0.7	2.0 ± 0.6	1.9 ± 0.6	0.35
HDL chol. mmol/L	1.4 ± 0.4	1.3 ± 0.4	1.2 ± 0.4	1.1 ± 0.3	<0.001
Triglycerides mmol/L	1.0 ± 0.4	1.2 ± 0.6	1.3 ± 0.6	1.5 ± 1.0	<0.001
Cod liver oil *n* (%)	144 (57.6%)	107 (43.0%)	92 (37.2%)	69 (27.8%)	<0.001
LA 18:2(*n*-6) %wt	19.3 ± 3.3	18.8 ± 3.0	18.3 ± 2.6	18.0 ± 2.5	<0.001
GLA 18:3(*n*-6) %wt	0.06 ± 0.03	0.07 ± 0.03	0.09 ± 0.03	0.10 ± 0.04	<0.001
AA 20:4(*n*-6) %wt	9.3 ± 2.3	10.3 ± 2.3	10.4 ± 2.2	10.3 ± 2.1	<0.001
LA:DGLA ratio	9.4 ± 2.7	7.0 ± 1.2	5.9 ± 0.9	4.7 ± 0.8	<0.001

Abbreviations: DGLA = dihomo-gamma (γ)-linolenic acid (20:3w6). BMI = Body Mass Index. SBP = Systolic Blood Pressure. LVEF = Left Ventricular Ejection Fraction. CKD = Chronic Kidney Disease. chol. = cholesterol. LDL = Low-Density Lipoprotein. HDL = High-Density Lipoprotein. LA = Linoleic Acid (LA). GLA = Gamma (γ)-Linolenic Acid (GLA). AA = Arachidonic Acid. *p*-Values for trends across the quartiles.

**Table 2 nutrients-13-03475-t002:** Baseline characteristics by quartiles (Q) of LA:DGLA.

	LA:DGLA	LA:DGLA	LA:DGLA	LA:DGLA Q 4	*p*-Value for Trend
	Q1	Q2	Q3	Q4	
No. of Patients	*n* = 251	*n* = 251	*n* = 251	*n* = 249	
*LA/DGLA ratio*	*<5.24*	*5.24–6.32*	*6.32–7.77*	*>7.77*	
Age years	74.4 ± 3.5	74.7 ± 3.6	74.6 ± 3.5	75.3 ± 3.7	*p* = 0.033
Sex (females) *n* (%)	87 (34.3%)	63 (24.8%)	68 (26.8%)	74 (29.2%)	*p* = 0.10
Smoking history					*p* = 0.64
Current smoker *n* (%)	30 (11.8%)	30 (11.8%)	29 (11.4%)	32 (12.6%)	
Previous smoker *n* (%)	123 (48.4%)	137 (53.9%)	117 (46.1%)	122 (48.2%)	
Never smoker *n* (%)	101 (39.8%)	87 (34.3%)	108 (42.5%)	99 (39.1%)	
BMI kg/m^2^	27.9 ± 4.4	27.5 ± 9.7	26.4 ± 3.8	25.3 ± 3.6	*p* < 0.001
SBP mmHg	137.9 ± 19.9	136.9 ± 20.4	136.1 ± 19.6	137.4 ± 19.1	*p* = 0.76
LVEF %	50.2 ± 8.4	50.3 ± 8.3	50.6 ± 8.8	48.4 ± 8.9	*p* = 0.10
Diabetes mellitu *n* (%)s	58 (22.8%)	47 (18.5%)	53 (20.9%)	53 (20.9%)	*p* = 0.69
Hyperlipidemia *n* (%)	116 (45.7%)	116 (45.7%)	115 (45.3%)	121 (47.8%)	*p* = 0.94
Hypertension *n* (%)	179 (70.5%)	148 (58.3%)	142 (55.9%)	143 (56.5%)	*p* = 0.002
CKD *n* (%)	8 (3.1%)	16 (6.3%)	8 (3.1%)	13 (5.1%)	*p* = 0.23
Heart failure *n* (%)	13 (5.1%)	23 (9.1%)	12 (4.7%)	17 (6.7%)	*p* = 0.18
LDL chol. mmol/L	1.9 ± 0.6	1.9 ± 0.6	2.0 ± 0.7	2.1 ± 0.7	*p* = 0.017
HDL chol. mmol/L	1.2 ± 0.3	1.2 ± 0.4	1.3 ± 0.4	1.4 ± 0.4	*p* < 0.001
Triglycerides mmol/L	1.5 ± 1.0	1.3 ± 0.6	1.2 ± 0.6	1.1 ± 0.5	*p* < 0.001
Cod liver oil *n* (%)	79 (31.5%)	97 (38.3%)	109 (43.3%)	131 (52.2%)	*p* < 0.001
LA 18:2(*n*-6) %wt	16.4 ± 2.2	17.8 ± 2.1	19.1 ± 2.3	21.0 ± 2.8	*p* < 0.001
GLA 18:3(*n*-6) %wt	0.11 ± 0.04	0.09 ± 0.03	0.07 ± 0.03	0.05 ± 0.02	*p* < 0.001
DGLA 20:3(*n*-6) %wt	3.7 ± 0.5	3.1 ± 0.4	2.7 ± 0.3	2.2 ± 0.4	*p* < 0.001
AA 20:4(*n*-6) %wt	10.9 ± 2.1	10.4 ± 2.2	10.1 ± 2.2	8.9 ± 2.1	*p* < 0.001

Abbreviations: LA = Linoleic acid. DGLA = dihomo-gamma (γ)-linolenic acid (20:3w6). BMI = Body Mass Index. SBP = Systolic Blood Pressure. LVEF = Left Ventricular Ejection Fraction. CKD = Chronic Kidney Disease. chol. = cholesterol. LDL = Low-Density Lipoprotein. HDL = High-Density Lipoprotein. LA = Linoleic Acid (LA). GLA = Gamma (γ)-Linolenic Acid (GLA). AA = Arachidonic Acid. suppl = supplements. *p*-Values for trends across the quartiles.

**Table 3 nutrients-13-03475-t003:** Endpoints/adverse events by quartiles of DGLA.

Total Pat. No.	DGLA Quartile 1	DGLA Quartile 2	DGLA Quartile 3	DGLA Quartile 4	*p*-Value for Trend
*n* = 1002	*n* = 251	*n* = 251	*n* = 251	*n* = 249	
MACE	61 (24.3%)	52 (20.7%)	46 (18.3%)	49 (19.7%)	0.16
AMI	18 (7.2%)	18 (7.2%)	18 (7.2%)	24 (9.6%)	0.33
Revasc.	16 (6.4%)	24 (9.6%)	16 (6.4%)	21 (8.4%)	0.69
Stroke	12 (4.8%)	9 (3.6%)	5 (2.0%)	6 (2.4%)	0.08
Heart failure	15 (6.0%)	9 (3.6%)	10 (4.0%)	11 (4.4%)	0.46
Total death	21 (8.4%)	12 (4.8%)	11 (4.4%)	11 (4.4%)	0.06

DGLA = dihomo-gamma (γ)-linolenic acid. MACE = Major Adverse Clinical Events (nonfatal MI, unscheduled coronary revascularization, stroke, all-cause death, or hospitalization for new or worsened heart failure). AMI = Acute Myocardial Infarction. Revasc. = Revascularization (percutaneous coronary intervention or coronary artery bypass grafting). HF = Heart Failure. suppl = supplements. *p*-Values for trends across the quartiles.

**Table 4 nutrients-13-03475-t004:** Univariate and Multivariable Cox regression model applying continuous values and comparison of Quartiles 2–4 (Q2–4) vs. Q1.

	DGLA		DGLA		LA:DGLA		LA:DGLA	
	MACE during 24 mo.		Total Death during 24 mo.		MACE during 24 mo.		Total Death during 24 mo.	
	HR(95% CI)	*p*-Value	HR (95% CI)	*p*-Value	HR (95% CI)	*p*-Value	HR (95% CI)	*p*-Value
**Univariate**								
Cont. values	0.89 (0.72–1.08)	0.24	0.72 (0.48–1.07)	0.11				
Q2–4 vs. Q1	0.76 (0.56–1.02)	0.067	0.51 (0.30–0.88)	0.016	1.42 (1.09–1.99)	0.046	1.73 (0.85–3.84)	0.13
**Multivarable**								
Model 1	0.78 (0.58–1.06)	0.12	0.54 (0.31–0.95)	0.03	1.34 (0.94–1.99)	0.11	1.65 (0.77–3.54)	0.20
Model 2	0.76 (0.56–1.04)	0.09	0.50 (0.28–0.91)	0.02	1.36 (0.95–1.94)	0.09	1.66 (0.77–3.58)	0.20
Model 3	0.75 (0.55–1.02)	0.07	0.47 (0.26–0.84)	0.012	1.42 (1.00–2.04)	0.052	1.80 (0.83–3.92)	0.14

Abbreviations: MACE = Major Adverse Clinical Events. Model 1: Adjusting for age, sex and BMI. Model 2: Adding cod liver oil supplementation at baseline to the model. Model 3: Adding hypertension, chronic kidney disease (CKD) and diabetes mellitus to the former models.

**Table 5 nutrients-13-03475-t005:** Endpoints/adverse events by quartiles of LA:DGLA.

Total Pat. No.	LA:DGLA Quartile 1	LA:DGLA Quartile 2	LA:DGLA Quartile 3	LA:DGLA Quartile 4	*p*-Value for Trend
*n* = 1002	*n* = 251	*n* = 250	*n* = 251	*n* = 250	
MACE	42 (16.7%)	50 (20.0%)	58 (23.1%)	58 (23.2%)	0.049
AMI	20 (8.0%)	21 (8.4%)	21 (8.4%)	16 (6.4%)	0.53
Revasc.	16 (6.4%)	20 (8.0%)	26 (10.4%)	15 (6.0%)	0.87
Stroke	6 (2.4%)	5 (2.0%)	8 (3.2%)	13 (5.2%)	0.05
Heart failure	9 (3.6%)	10 (4.0%)	14 (5.6%)	12 (4.8%)	0.37
Total death	10 (4.0%)	14 (5.6%)	11 (4.4%)	20 (8.0%)	0.09

LA = linoleic acid. MACE = Major Adverse Clinical Events (nonfatal MI, unscheduled coronary revascularization, stroke, all-cause death or hospitalization for new or worsened heart failure). AMI = Acute Myocardial Infarction. Revasc. = Revascularization (percutaneous coronary intervention or coronary artery bypass grafting). HF = Heart Failure. *p*-values for trends across the quartiles.

## Data Availability

Data supporting the reported study results are stored in a local database.

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
