# Peer review of "Serum Levels of Dihomo-Gamma (γ)-Linolenic Acid (DGLA) Are Inversely Associated with Linoleic Acid and Total Death in Elderly Patients with a Recent Myocardial Infarction"

_nutrients, 2021, doi:10.3390/nu13103475_

Round 1
Reviewer 1 Report
To authors
In this paper, the authors aimed to examine the impact of serum dihomo-gamma-linolenic acid (DGLA) and linoleic acid (LA):DGLA on the all-cause mortality and major cardiovascular events. To date, such an issue has not been well established with diverse results in previous literature. In this context, I think their findings are somewhat interesting, but several concerns are listed below.
1) In my opinion, there is some gap between the purpose to the results. The main outcome was all-cause mortality, and cardiovascular outcomes would be a secondary outcomes. However, in the abstract, there is no statement for mortality in the aim section.
2) I think the approach to drawing the result was a critical weak point in this study. Elderly patients have lots of risk factors for mortality, and the author demonstrated these factors in the baseline characteristics. However, in the Cox-proportional analysis, the author used limited variables such as age, sex, BMI, intake of cod liver oil, and hypertension. Except these variables, diabetes and CKD status could be significant factors for mortality and could be confounding factors, so it should be added to the model.
3) The author used DGLA and other fatty acids variables corrected by % weight. However, this variable is strongly influenced by the nutritional state and diet intake. Therefore, it would be better to use %energy value, if there is a variable for the total calorie intake.
4) I wonder there is data for the laboratory results. If there are some variables representing the nutritional status, such as albumin, cholesterol, then it would be better to use for adjustment in the analysis.
5) Based on the result, mortality was significantly associated with lower level of DGLA. However, in the K-M curve, there was no linear relationship, and I guess it could be U-shaped relationship.
Instead of the supplemental Figure 3, please show the spline curve for DLGA and mortality. Moreover, it would be helpful the threshold for the beneficial or harmful value.
6) There were incorrectly cited tables in the result section.
* Result 4.1. DGLA as predictor of adverse outcome – “These associations remained essentially unchanged and and statistically non-specificant in the multivariable Cox regression models (Table 4)”
> Maybe it is incorrectly cited. (Table 4 might be Table 5)
* Result 4.2. LA:DGLA ratio as predictor of outcomes – “The HR for Q2-4 combined as compared to Q1 was significant in the univariate analysis [1.42 (1.09 – 1.99), p=0.046], but became borderline significant after adjusting (Table 4) for the variables in Model 3 [1.42 (0.99-2.03)], p=0.055].”
> Maybe it is incorrectly cited. (Table 4 might be Table 5)
Reviewer 2 Report
Nilsen DWT et al. investigated the correlation of the n-6 fatty acid DGLA and the LA:DGLA ratio – an indirect marker of conversion of LA to DGLA and thus delta-6-desaturase activity – with MACE (a composite of nonfatal MI, unscheduled coronary revascularization, stroke, hospitalization for heart failure or all-cause death) and overall death in 1002 patients after myocardial infarction (a substudy of the recent OMEMI trial). The authors found no significant association of DGLA quartiles with MACE or with death. Also, a continuous analysis showed no significant association of DGLA with MACE or with death. Finally, the authors found that the highest quartiles of DGLA (i.e. quartile 2-4), compared with quartile 1 were associated with a lower risk for the development of death, but not MACE. This findings suggest that low levels of DGLA may be deleterious whereas higher levels of DGLA may have no additional beneficial value. The study is in line with previous findings of Nilsen DWT et al. (Int J Cardiol. 2017) and expands the current evidence of DGLA and its prognostic potential for MACE and total mortality in patients after ACS.
Yet, the study has major limitation
- It is surprising that omega-3 FA supplementation is inversely associated with DGLA quartiles at baseline. It is expected to be similar in all groups. I suggest that with “omega-3 FA supplementation” the authors refer to the omega-3 FA supplementation from the OMEMI trial (i.e. the intervention group) and not the cod liver supplementation, which was used by only 20% of study participants. When was the blood for DGLA analysis drawn – before or after beginning of omega-3 FA supplementation? If blood was drawn after the beginning of supplementation, it appears as if omega-3 FAs (or corn oil) affect DGLA levels, which would introduce a bias for the analysis over time as DGLA levels are changing. In order to exclude the bias of omega-3 fatty supplementation on DGLA levels, the authors should perform a sensitivity analysis in participants without omega-3 fatty acid supplementation. The other explanation would be that corn oil (rich in LA) affects DGLA levels, however patients with high levels of DGLA have lower levels of LA at baseline. Please also list corn oil supplementation in the base line characteristics.
- The authors performed 3 models. Model 1 was adjusted for age, sex and BMI, model 2 was also adjusted for baseline omega-3 FA supplementation and model 3 for hypertension. This adjustments appear to be incomplete. The authors should list and correct for all risk factors that could affect their primary outcome including smoking, HDL/LDL/TAG levels, kidney disease, diabetes, heart failure, atrial fibrillation, dyslipidemia, antiplatelet therapy, dual antiplatelet therapy, anticoagulation, index type of myocardial infarction and revascularization strategy.
Minor comments
- What was the mean follow up?
- Please describe the range of DGLA quartiles in addition to mean +/- SD
Reviewer 3 Report
This study was conducted to evaluate the association between DGLA and LA:DGLA and the development of cardiovascular events.
The study design is a very well set up design.
I believe that the methods and statistical treatment are appropriate.
Author Response
Please see attchment.

Round 2
Reviewer 1 Report
Thank you for the sincere response to the comment.
All of the comments were well responded to, and I have no more comment.